# Improving Access to Sexual Health Services in General Practice Using a Hub-and-Spoke Model: A Mixed-Methods Evaluation

**DOI:** 10.3390/ijerph19073935

**Published:** 2022-03-25

**Authors:** Jason J. Ong, Christopher K. Fairley, Ria Fortune, Melanie Bissessor, Chantal Maloney, Henrietta Williams, Adrian Castro, Lea Castro, Jason Wu, Pei Sue Lee, Eric P. F. Chow, Marcus Y. Chen

**Affiliations:** 1Melbourne Sexual Health Centre, Alfred Health, Melbourne 3053, Australia; cfairley@mshc.org.au (C.K.F.); rfortune@mshc.org.au (R.F.); mbissessor@mshc.org.au (M.B.); cmaloney@mshc.org.au (C.M.); hwilliams@mshc.org.au (H.W.); echow@mshc.org.au (E.P.F.C.); mchen@mshc.org.au (M.Y.C.); 2Central Clinical School, Monash University, Melbourne 3004, Australia; 3Kings Park Medical Centre Hillside, Hillside 3037, Australia; drcastro@bigpond.com (A.C.); leacastro31@gmail.com (L.C.); jasun_quo@hotmail.com (J.W.); 4Tarneit Family Medical & Dental Centre, Tarneit 3029, Australia; leepeisue@hotmail.com

**Keywords:** HIV, sexually transmitted infection, general practice, hub and spoke, primary care, sexual health

## Abstract

Improving access to sexual health services is critical in light of rising sexually transmitted infections (STIs). We evaluated a hub-and-spoke model for improving access to sexual health services in three general practices in Victoria, Australia. The primary outcome was the impact on HIV and STI (chlamydia, gonorrhoea, syphilis) testing. Segmented linear regression analysis was conducted to examine the trends in the total HIV/STI tests pre- (from January 2019 to June 2020) and post-implementation (from July 2020 to July 2021). We evaluated the feasibility and acceptability of integrating this model into the general practices using semi-structured individual interviews. There was a statistically significant rise in testing for HIV and STIs in all general practices: post-implementation, there was an increase of an average of 11.2 chlamydia tests per month (*p* = 0.026), 10.5 gonorrhoea tests per month (*p* = 0.001), 4.3 syphilis tests per month (*p* = 0.010), and 5.6 HIV tests per month (*p* = 0.010). Participants reported increases in knowledge level and confidence in offering STI testing and managing a greater variety of sexual health cases. This study demonstrates the feasibility of implementing a hub-and-spoke model to enable GPs to deliver sexual health care with support from a sexual health specialist service.

## 1. Introduction

The World Health Organization estimates more than a million new sexually transmitted infections (STIs) a day globally [1]. In Australia, there are rising rates of chlamydia, gonorrhoea, and syphilis [2]. Of particular concern is the rise in congenital syphilis, which more than doubled from 7.8 per 100,000 in 2013 to 18.3 per 100,000 in 2017 [2]. There are also significant social (stigma and discrimination) and economic consequences for HIV/STI (US$16.7 billion (range 11.8–22.1); 3.2 billion without considering HIV) [3]. In recent years, there have been reports of rising antimicrobial resistance for *Neisseria gonorrhoeae* and *Mycoplasma genitalium* globally [4].

To control HIV/STI, healthcare systems must support earlier testing and management because a shorter duration of infectiousness has a powerful effect in reducing the incidence of HIV/STI [5]. Most STIs are treatable, and earlier detection and management can reduce health consequences such as reproductive morbidity (e.g., infertility, abortion, preterm birth), cancer (e.g., cervical cancer from HPV), HIV transmission, and lifelong disability or death (e.g., from congenital syphilis or herpes). There is considerable evidence that HIV/STI are driven primarily by lack of access to services rather than sexual risk behaviours. For example, in the United States, there is a lifetime HIV risk of 1 in 22 for black men and 1 in 122 for white men [6]. This large discrepancy is not primarily due to differences in condom use and partner numbers but is driven by the reduced access of black men to HIV testing, treatment, and pre-exposure prophylaxis (PrEP). Similarly, in Australia, though there is a reduction in HIV incidence among Australian-born gay, bisexual, and other men who have sex with men (GBM), there is no change in HIV incidence among overseas-born GBM [7]. This difference may be explained by reduced access to HIV testing, treatment, and PrEP for Medicare-ineligible patients, as recently arrived GBM have fewer sexual partners than Australian-born GBM [8]. Medicare is Australia’s public health insurance system to provide subsidised healthcare to citizens and permanent residents. Another powerful example of the impact of access to services on infectious diseases was the widespread access to penicillin after the end of World War II in the mid-1940s, with associated dramatic decreases in syphilis [9].

General practitioners play a critical role in improving HIV/STI testing access. In Australia, it is estimated that 83% of the general population would visit a general practitioner in a year [10]. There have been several attempts at improving HIV/STI testing among GPs with mixed success [11]. The perceived lack of sexual health expertise from GPs is a barrier [12]. However, there are successes in the Australian HIV s100 prescriber system, where people living with HIV receive ongoing management for HIV (including regular STI testing) within general practice [13]. Although the provision of specialised services in primary care has been adopted using a hub-and-spoke model to improve access to general medical services in resource limited settings [14], there have not been any published examples of providing sexual health services using a hub-and-spoke model.

The hub-and-spoke model is a network with a central facility that provides a full array of sexual health primary and specialist services (“the hub”) supporting geographically dispersed secondary services (“the spokes”), which provides primary but more limited specialist services [15]. This arrangement allows for less complicated patients to be primarily served by the spokes and more complex patients to be managed at the spokes with active support or redirected to the hub when necessary. This active collaboration between the hub and its spokes could facilitate greater consistency across services in terms of efficiencies, quality of care, and enhanced accessibility of specialised services for the community. This model may be more efficient than independently replicating multiple specialist services across a geographical area and is more easily scalable (i.e., adding additional spokes when needed).

This study aimed to evaluate a hub-and-spoke model for improving access and increasing HIV/STI testing in three general practices in Victoria, Australia. The primary outcome was to evaluate the impact on HIV and STI (chlamydia, gonorrhoea, syphilis) testing. We assessed the feasibility and acceptability of integrating this model into the general practices using qualitative methods.

## 2. Materials and Methods

In 2020, Melbourne Sexual Health Centre (MSHC), a part of Alfred Health, undertook a project funded by the Victorian Department of Health and Human Services in response to the Victorian Review of Sexual Health Services, which recommended the development of HIV/STI services provided in a decentralised hub-and-spoke model where Melbourne Sexual Health Centre served as the specialist hub for HIV/STI testing and treatment supporting GP spokes that provide primary care HIV/STI testing and treatment. The training phase of the project took place between May 2020 and July 2020. The support Melbourne Sexual Health Centre delivered to these three GP partner services is provided in the Appendix A. Three suburban GP practices were identified and included in the study located west of Melbourne (Clinic 1), south of Melbourne (Clinic 2), and north-west of Melbourne (Clinic 3). These locations were selected based on syphilis prevalence data that syphilis infection has spread to outer metropolitan areas of Melbourne [16]. We used a mixed-methods approach: extracting quantitative data on HIV/STI testing complemented by qualitative interviews to evaluate the acceptability and feasibility of integrating the hub-and-spoke model into each GP.

We defined post-implementation in our study as after the month when clinics started seeing patients using the GP hub-and-spoke model. We extracted data from the pathology testing laboratory on the number of tests (HIV, syphilis, chlamydia, gonorrhoea) and test positivity for each clinic from two time periods: (1) pre-implementation phase: January 2019 to June 2020; and (2) post-implementation phase: July 2020 to July 2021. Descriptive statistics were used to analyse the data. We used Student’s *t*-test to compare the average number of tests for each pathogen pre-implementation and implementation phase. Segmented linear regression analysis was conducted to examine the trends in the total HIV/STI tests in pre- (from January 2019 to June 2020) and post-implementation phases (from July 2020 to July 2021).

A total of 33 semi-structured individual interviews were conducted from May to June 2020, before the training: 5 from Clinic 1, 13 from Clinic 2, and 15 from Clinic 3. Interviews explored the staff’s current knowledge, attitude, and practice related to sexual health; current challenges in providing sexual health care in the general practice; and training needs. A second interview was conducted among staff who had experience providing the sexual health services in the respective clinics in August and September 2020. Interviews explored their experiences in delivering sexual health care as part of the hub-and-spoke model; feedback on the training received for the implementation of the sexual health service; changes in their knowledge, attitude, and practice as a result of participating in the training; and ongoing support needs. The interview guide is provided in the Appendix A. All interviews were recorded and data analysed using a content analysis approach to summarise the main themes [17]. A researcher (JO) identified initial themes, and reviewed and refined the themes with the research team. Transcripts were then re-read, and noteworthy phrases or concepts were coded and categorised into themes and sub-themes derived both deductively from the interview schedule and the data. No financial reimbursement was offered to interviewees.

As this was part of a quality improvement and evaluation activity, we received a waiver from the Alfred Hospital Human Research Ethics Committee. A researcher (JO) approached each of the GP clinics’ practice managers to approach their clinic staff explaining the voluntary nature of the evaluation, and interviews were only conducted with those who agreed to participate. Consent was implied when people participated.

## 3. Results

Table 1 summarises the average number of HIV/STI tests per GP clinic, pre-and post-training. It demonstrates a significant increase in testing for chlamydia, gonorrhoea, syphilis, and HIV across all three general practices. Pre-implementation, there was no significant change in the mean monthly chlamydia tests (*p* = 0.246, Figure 1). Post-implementation, there was an immediate increase in the number of chlamydia tests of 33.7 in the first month after the GP hub-and-spoke was launched. This was followed by a significant increase of an average of 11.2 tests per month (*p* = 0.026).

Pre-implementation, there was no significant change in the mean monthly gonorrhoea tests (*p* = 0.826, Figure 2). Post-implementation, there was an immediate increase in the number of gonorrhoea tests of 38.4 in the first month after the GP hub-and-spoke was launched. This was followed by a significant increase of an average of 10.5 tests per month (*p* = 0.001).

Pre-implementation, there was no significant change in the mean monthly syphilis tests (*p* = 0.122, Figure 3). Post-implementation, there was an immediate increase in the number of syphilis tests of 27.0 in the first month after the GP hub-and-spoke was launched. This was followed by a significant increase of an average of 4.3 tests per month (*p* = 0.010).

Pre-implementation, there was a significant decrease in HIV tests of 2.3 per month (*p* = 0.002, Figure 4). Post-implementation, there was an immediate increase in HIV tests of 30.4 in the first month after the GP hub-and-spoke was launched. This was followed by a significant increase of an average of 5.6 tests per month (*p* = 0.010).

There was no significant change in test positivity for chlamydia, gonorrhoea, syphilis, and HIV (Appendix A). Results from the qualitative interviews are presented in the Appendix A. In brief, prior to training, there was low knowledge about specific aspects of sexual health, positive attitude towards sexual minorities, wanting to learn more about sexual health, and low confidence in managing certain types of cases, e.g., most GPs referred cases related to HIV pre-exposure prophylaxis (PrEP), HIV post-exposure prophylaxis (PEP), and syphilis to Melbourne Sexual Health Centre for specialist management. GP staff expressed their training needs and challenges in providing sexual health care in a general practice setting. Following training, all clinics expressed increased knowledge and confidence in offering sexual health services, an increased interest in sexual health, and increased attractiveness of their general practice for recruitment of new staff. Overall, study participants reported that the hub-and-spoke approach was highly acceptable and that integration of the model into their current workflows and business operation made the model highly feasible.

## 4. Discussion

We evaluated a program that implemented a hub-and-spoke method to deliver sexual health services in Victoria, Australia. We found that the hub-and-spoke model led to an early and sustained increase in chlamydia, gonorrhoea, syphilis, and HIV testing. This is despite the social restrictions imposed in Victoria because of the COVID-19 pandemic. Our study adds to the scant literature on how a hub-and-spoke model can be used to improve access to sexual health services [18].

A key target of the Fourth Australian National Sexually Transmissible Infections Strategy (2018–2022) is to increase STI testing coverage [5]. Enabling timely access to testing and treatment requires new strategies and policies to enable accessible STI care. There are several ways to configure HIV/STI testing and management services, including the potential for integrating novel service delivery models such as virtual STI clinics [19], postal services [20], or pharmacy-based services [21]. However, decision-makers have limited evidence on which programs are likely to be used by those at greatest need, which are most cost-effective, and which should be scaled up. Our findings demonstrate that the GP hub-and-spoke model significantly increased and sustained testing (at least for one year) for chlamydia, gonorrhoea, syphilis, and HIV. This is despite the restrictions related to the COVID-19 pandemic in Victoria.

There are several advantages to the GP hub-and-spoke model. Strengthening general practices to provide sexual health care can help destigmatise sexual health services by providing them through GPs who are already accessible and trusted by the community. Patients who book in for a sexual health consultation with their GP may feel more comfortable discussing sexual health matters, knowing their GP is interested and skilled in providing this care [22]. Normalisation and destigmatisation of STI testing are critical for reducing the burden of STI and promoting sexual health. The hub-and-spoke model could also be a more efficient use of specialist services, i.e., complex cases from spokes can be redirected to the hub and vice versa—send “basic” cases to the spokes. Given that most of the population encounter a GP every year (83.2% in 2019/2020) [10], there is an opportunity for opportunistic screening when people present with non-sexual-health-related complaints or issues—reaching people who may not attend public sexual health services. In addition, GPs can offer additional services such as vaccination, prevention (PrEP), counselling, mental health, and substance-use support for those who need these services. Scaling up the hub-and-spoke model, particularly to remote and rural areas, can increase the accessibility of sexual health services to these underserved populations.

Our model of care uses the nurse to perform the initial assessment. This is beneficial for embedding STI testing in the practice, creating extra capacity for patient appointments, and workforce development for sexual health nurses [23]. Our study also identified several challenges in creating a hub-and-spoke model for sexual health services. First, it is not clear what are the predictors or markers of a successful spoke. In our experience thus far, we found that three elements were important: (1) commitment and involvement of the principal GP; (2) one or more staff having a particular interest in delivering sexual health care to their patients; and (3) the governance structure of the practice/service meaning that practice staff are consulted, involved, and supported to participate in delivering sexual health care. This is critical, as the training time and resource investment are significant (Appendix A). Second, building trust and relationships with the local communities to create a safe space for sexual health services will take time. This includes appropriate and targeted marketing. Unlike traditional hub-and-spoke organisational designs [14], we allowed spokes to have relative independence in adapting the sexual health service to the GPs’ needs. This may lead to inconsistency across operations but functionally, it was more suitable to local needs. Our study was not a hierarchical structure with authority of hubs over the spokes. This means that patients may not have similar experiences across all services (not like a franchise). Therefore, there may be some risks related to unmet expectations of patients expecting the same level of service/expertise when attending a specialist service, with the potential to tarnish the reputation of the organisations if a high-quality service is not sustained. Mitigation of this risk requires ongoing quality assurance and support. Third, there may be direct impacts on the patient flow in the GP practice. For example, patients would initially see the nurse (task-shifting) upskilled to provide sexual health care. Fourth, if there is a high staff turnover in the GP clinic, this necessitates frequent and ongoing training for new staff.

Our results must be read in light of some limitations. The evaluation was conducted during major disruptions to primary care services because of COVID-19-related social distancing restrictions. General practices during this period had to prioritise COVID-19-related activities (including delivering vaccines) and transition from in-person to telehealth services. Nevertheless, we still observed a significant rise in HIV/STI testing post-implementation. However, we did not observe any impact on test positivity, which could either reflect a lower incidence of STI because of declines in sexual activity [24] or an increase in testing among low-risk populations. There will be ongoing monitoring of HIV/STI testing and positivity rates as the hub-and-spoke model is scaled up in Victoria. In addition, collecting data related to age, gender, and sexual orientation would be helpful in generalizing our results to other settings.

## 5. Conclusions

This study demonstrated that GPs are well placed to deliver sexual health care with specialist services providing support and training through a hub-and-spoke model. This ongoing and synergistic collaboration has led to a sustained increase in HIV and STI testing. Further implementation research is warranted to evaluate its scalability in other settings and for managing other conditions.

## Figures and Tables

**Figure 1 ijerph-19-03935-f001:**
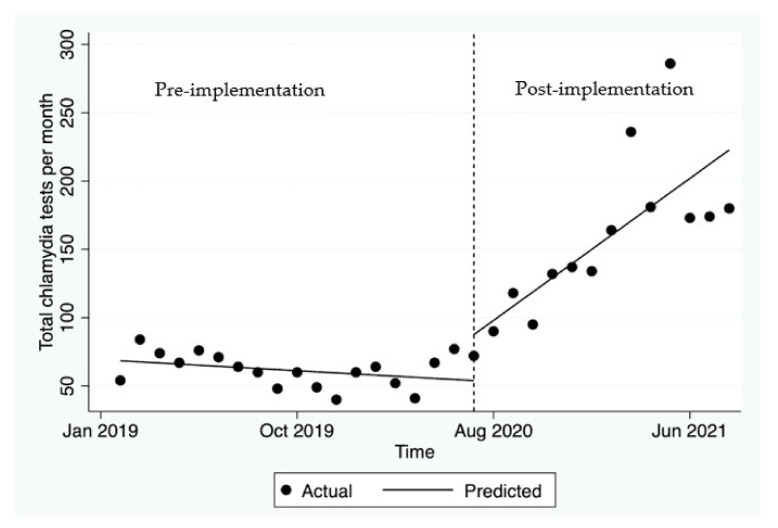
Number of tests for chlamydia before and after the hub-and-spoke model was launched.

**Figure 2 ijerph-19-03935-f002:**
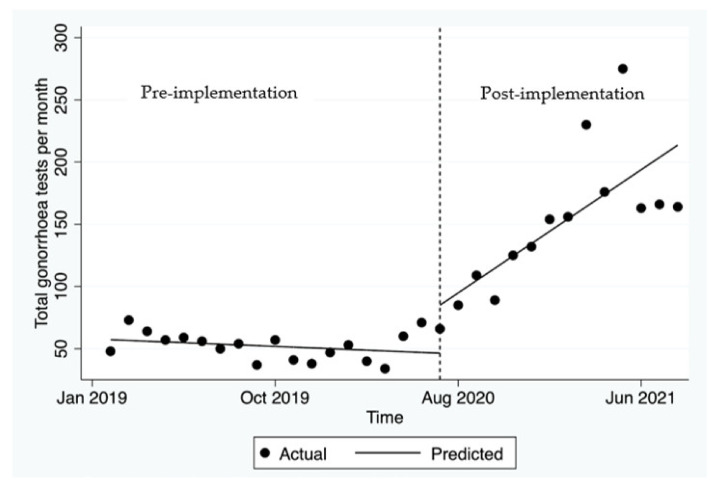
Number of tests for gonorrhoea before and after the hub-and-spoke model was launched.

**Figure 3 ijerph-19-03935-f003:**
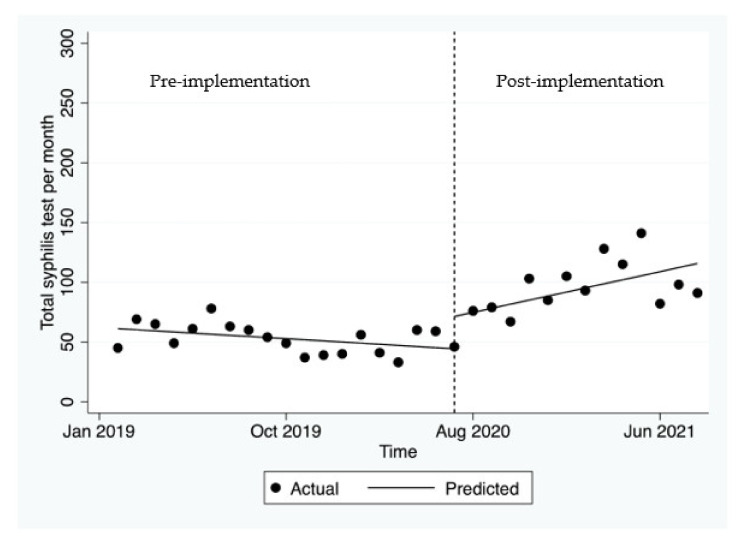
Number of tests for syphilis before and after the hub-and-spoke model was launched.

**Figure 4 ijerph-19-03935-f004:**
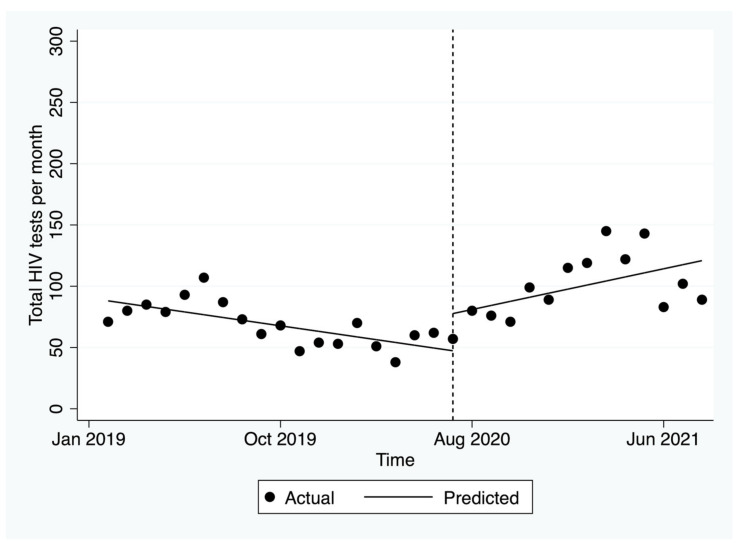
Number of tests for HIV before and after the hub-and-spoke model.

**Table 1 ijerph-19-03935-t001:** The average number of HIV/STI tests.

	Before Implementation—Average Number per Month (SD)	After Implementation—Average Number per Month (SD)	*p*-Value
**Chlamydia**			
Clinic 1	16.7 (5.6)	35.5 (15.5)	<0.0001
Clinic 2	13.1 (4.3)	55.5 (14.8)	<0.0001
Clinic 3	31.8 (6.7)	64.1 (37.4)	<0.0001
Total	61.6 (12.5)	155.1 (57.6)	<0.0001
**Gonorrhoea**			
Clinic 1	16.0 (5.6)	34.9 (13.8)	<0.0001
Clinic 2	11.8 (3.8)	51.1 (12.5)	<0.0001
Clinic 3	24.3 (5.8)	63.3 (38.1)	<0.0001
Total	52.2 (11.3)	114.5 (33.0)	<0.0001
**Syphilis**			
Clinic 1	20.0 (5.2)	24.3 (7.9)	<0.0001
Clinic 2	12.7 (5.9)	33.7 (7.4)	<0.0001
Clinic 3	20.5 (7.1)	35.5 (16.1)	<0.0001
Total	53.2 (12.4)	93.5 (24.6)	<0.0001
**HIV**			
Clinic 1	21.1 (5.0)	24.6 (7.9)	<0.0001
Clinic 2	23.3 (6.6)	34.3 (7.3)	<0.0001
Clinic 3	24.4 (11.6)	40.4 (18.1)	<0.0001
Total	68.8 (17.6)	99.3 (26.5)	<0.0001

SD = standard deviation.

## Data Availability

All relevant data are presented in this manuscript. Further details can be obtained by contacting the corresponding author.

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
