# Peer review of "Improving Access to Sexual Health Services in General Practice Using a Hub-and-Spoke Model: A Mixed-Methods Evaluation"

_ijerph, 2022, doi:10.3390/ijerph19073935_

Round 1
Reviewer 1 Report
In the manuscript “Improving access to sexual health services in general practice using a Hub-and-spoke model: a mixed-methods evaluation”, Ong and colleagues suggest an excellent model to improve access to sexual health services. The model is simple, low cost, and relatively easy to be implemented. The work is deeply relevant once models to increase access to testing and public education policy targeting STIs are urgent. In the manuscript, the sample group should be expanded and better fitted to the population, however, the data look promising.
Author Response
Reviewer 1
In the manuscript “Improving access to sexual health services in general practice using a Hub-and-spoke model: a mixed-methods evaluation”, Ong and colleagues suggest an excellent model to improve access to sexual health services. The model is simple, low cost, and relatively easy to be implemented. The work is deeply relevant once models to increase access to testing and public education policy targeting STIs are urgent. In the manuscript, the sample group should be expanded and better fitted to the population, however, the data look promising.
Response:
We thank the reviewer for their careful review and positive comments about our research. We agree that providing further details of the study population’s demographics would be helpful in generalizing our findings to the rest of the population. Unfortunately, the current data extracted was anonymous and did not have details related to gender, age, and sexual orientation. Further implementation research is planned as we expand this model in our state (e.g. 3 more general practices have been added recently using this GP hub and spoke approach since the writing of our manuscript).
Changes:
(Line 294)
In addition, collecting data related to the age, gender and sexual orientation would be helpful in generalizing our results to other settings.
Reviewer 2 Report
In this study, Ong and co-authors have assessed the feasibility and acceptability of a Hub and Spoke model into the general practices for improving access to sexual health services in Victoria, Australia. Significant rise in testing for HIV and STIs was observed in the participated clinics.
Overall the findings are important towards the improvement of access to sexual health services among the targeted populations. However, there are major drawbacks that should be addressed adequately in a revised version. Major and specific comments are given below.
MAJOR REVISION
- The study aimed to assess the feasibility and acceptability of integrating this model into the general practices. It is mentioned that the primary outcome was the impact on HIV and STI (chlamydia, gonorrhoea, syphilis) testing. This outcome variable is useful to assess the impact of the model but not for the feasibility and acceptability.
My major concern here is about what part in results section indicated the feasibility and acceptability of the model? The results were on the changes in testing that reflect the impact or effectiveness of the model. It might be used to judge acceptability (this should be clearly indicate and discuss); but what about the feasibility?
- The study tools (questionnaires) should be adequately described in methods section. The authors mentioned assessment of participant’s knowledge, attitude and practice related to sexual health, and their feedback on training and the needs; thus, information on the questionnaires and interviews approaches is essential. Moreover, information about the training conditions and who have provided the training should also be provided.
- Lines 117-118: As this was part of a quality improvement and evaluation activity, we received a waiver from the Alfred Hospital Human Research Ethics Committee. This statement can be deleted. It is mentioned in the declaration section (please add ethical committee name).
I have a concern about this issue. The waiver of ethical approval for this study may be given concerning the patients as indicated in “Informed Consent Statement”, and this is acceptable. But, there is still a need for ethical approval and informed consent from the clinics’ staff participated in the evaluation of the model. The authors should confirm this point.
- Results about participant’s knowledge, attitude and practices related to sexual health can be presented in brief; at least the main changes in their KAP after the training and implementation of the model.
SPECIFIC COMMENTS
- The training phase of the project took place between May 2020 and July 2020. However, in methods section, it is mentioned few twice that pre- implementation was from January 2019 to June 2020 and post-implementation was from July 2020 to July 2021. These periods involved the May to July training period. This should be amended.
- Line 64: “A hub-and-spoke model has been adopted for other diseases”. Give examples of those diseases.
- Line 101: correct to “and post-implementation phase.”
- Implementation between July 2020-February 2021 (line 99) and also mention as July 2020 to July 2021 (line 103).
- Lines 118-123: These statements about funding and availability of data can be removed. They are mentioned in declaration section.
- Lines 127-128: “Pre-implementation, there was no significant change in the total chlamydia tests (p=0.246, Figure 1).” What was the period? Do you mean monthly change (per month)? Please indicate. This comment is also applied on line 140 and line 149.
- Figures 1, 2 and 3: label both phases on the figures (i.e. pre and post-implementation).
- Training was provided between May and July 2020; but this period was also included in Figures 1-3.
- Supplementary file: Supplementary Table 1 summarises the impact of the GP hub and spoke model before and after implementation. Impact on what? Please indicate.
- STI or STIs. Please ensure consistency throughout the manuscript and supplementary file.
- Line 204: “Our model of care uses the nurse to do the initial assessment.” This is only mentioned here. This and additional important information about the model design, components and evaluation should be provided in methods section.
- Conclusions: A statement to recommend further evaluation can also be added here.
- Title: “a mixed-methods evaluation”. Nothing is mentioned in the text about the mixed methods. What does this mean?
Author Response
Reviewer 2, Comment 1
In this study, Ong and co-authors have assessed the feasibility and acceptability of a Hub and Spoke model into the general practices for improving access to sexual health services in Victoria, Australia. Significant rise in testing for HIV and STIs was observed in the participated clinics. Overall the findings are important towards the improvement of access to sexual health services among the targeted populations. However, there are major drawbacks that should be addressed adequately in a revised version. Major and specific comments are given below.
MAJOR REVISION
The study aimed to assess the feasibility and acceptability of integrating this model into the general practices. It is mentioned that the primary outcome was the impact on HIV and STI (chlamydia, gonorrhoea, syphilis) testing. This outcome variable is useful to assess the impact of the model but not for the feasibility and acceptability.
My major concern here is about what part in results section indicated the feasibility and acceptability of the model? The results were on the changes in testing that reflect the impact or effectiveness of the model. It might be used to judge acceptability (this should be clearly indicate and discuss); but what about the feasibility?
Response: The feasibility and acceptability of the project was based on the semi-qualitative interviews. This data is presented in Appendix 3. We have added further details in the results section to specifically address the feasibility and acceptability of the model.
Changes:
(Line 205)
Results from the qualitative interviews are presented in Appendix 3. In brief, prior to training, there was low knowledge about specific aspects of sexual health, positive attitude towards sexual minorities, wanting to learn more about sexual health, and low confidence in managing certain types of cases e.g. most GPs referred cases related to HIV pre-exposure prophylaxis (PrEP), HIV post-exposure prophylaxis (PEP) and syphilis to Melbourne Sexual Health Centre for specialist management. GP staff expressed their training needs and challenges in providing sexual health care in a general practice setting. Following training, all clinics expressed increased knowledge and confidence in offering sexual health services, an increased interest in sexual health, and increased attractiveness of their general practice for recruitment of new staff. Overall, study participants reported that the hub-and-spoke approach was highly acceptable and that integration of the model into their current workflows and business operation made the model highly feasible.
Reviewer 2, Comment 2
The study tools (questionnaires) should be adequately described in methods section. The authors mentioned assessment of participant’s knowledge, attitude and practice related to sexual health, and their feedback on training and the needs; thus, information on the questionnaires and interviews approaches is essential. Moreover, information about the training conditions and who have provided the training should also be provided.
Response: We have added the interview guide for the qualitative interviews in Appendix 2. We have added more information about training conditions and who provided the training in Appendix 3.
Changes:
(Line 129)
The interview guide is provided in Appendix 2.
(Line 376)
Training and education to GP clinic staff
Melbourne Sexual Health Centre assessed each of the clinics to identify their needs. Training and education were provided through face-to-face teaching in the GP clinics, teleconferencing, and visits to Melbourne Sexual Health Centre by GP and nursing staff where they observed patient consultations undertaken by sexual health physicians and nurses. Administrative staff from each GP also visited Melbourne Sexual Health Centre to observe the administrative operations.
Reviewer 2, Comment 3
Lines 117-118: As this was part of a quality improvement and evaluation activity, we received a waiver from the Alfred Hospital Human Research Ethics Committee. This statement can be deleted. It is mentioned in the declaration section (please add ethical committee name).
I have a concern about this issue. The waiver of ethical approval for this study may be given concerning the patients as indicated in “Informed Consent Statement”, and this is acceptable. But, there is still a need for ethical approval and informed consent from the clinics’ staff participated in the evaluation of the model. The authors should confirm this point.
Response: We deleted the sentence as suggested by the Reviewer. Regarding the waiver, we discussed this with our local ethics committee and no formal consent form was needed by the clinic staff participating in the evaluation of the model. Our research ethics committee does not require ethical approval for evaluation of new services. We have added further details to explain this.
Changes:
(Line 137)
A researcher (JO) approached each of the GP clinic’s practice manager to approach their clinic staff explaining the voluntary nature of the evaluation, and interviews were only conducted with those who agreed to participate. Consent was implied when people participated.
Reviewer 2, Comment 4
Results about participant’s knowledge, attitude and practices related to sexual health can be presented in brief; at least the main changes in their KAP after the training and implementation of the model.
Response: We agree and have added a summary in the main text.
Changes:
(Line 205)
Results from the qualitative interviews are presented in Appendix 3. In brief, prior to training, there was low knowledge about specific aspects of sexual health, positive attitude towards sexual minorities, wanting to learn more about sexual health, and low confidence in managing certain types of cases e.g. most GPs referred cases related to HIV pre-exposure prophylaxis (PrEP), HIV post-exposure prophylaxis (PEP) and syphilis to Melbourne Sexual Health Centre for specialist management. GP staff expressed their training needs and challenges in providing sexual health care in a general practice setting. Following training, all clinics expressed increased knowledge and confidence in offering sexual health services, an increased interest in sexual health, and increased attractiveness of their general practice for recruitment of new staff. Overall, study participants reported that the hub-and-spoke approach was highly acceptable and that integration of the model into their current workflows and business operation made the model highly feasible.
Reviewer 2, Comment 5
SPECIFIC COMMENTS
The training phase of the project took place between May 2020 and July 2020. However, in methods section, it is mentioned few twice that pre- implementation was from January 2019 to June 2020 and post-implementation was from July 2020 to July 2021. These periods involved the May to July training period. This should be amended.
Response: We apologize for the confusion. The pre-implementation phase was a retrospective evaluation that included data from January 2019 to June 2020 from the pathology testing laboratory, whereas the training phase was when the program actually begun. We have clarified this further in the text.
Changes:
(Line 110)
We defined post-implementation in our study as after the month when clinics started seeing patients using the GP hub-and-spoke model.
Reviewer 2, Comment 6
Line 64: “A hub-and-spoke model has been adopted for other diseases”. Give examples of those diseases.
Response: The reference we provided did not specify a specific disease but discusses how this model can be adopted in resource limited settings to improve access to medical care. We have re-phrased the sentence.
Changes:
(Line 71)
Although the provision of specialised services in primary care has been adopted using a hub-and-spoke model to improve access to general medical services in resource limited settings,[14] there have not been any published examples of providing sexual health services using a hub-and-spoke model.
Reviewer 2, Comment 7
Line 101: correct to “and post-implementation phase.”
Changes:
(Line 116)
Segmented linear regression analysis was conducted to examine the trends in the total HIV/STI tests before (from January 2019 to June 2020) and post-implementation phases (from July 2020 to July 2021).
Reviewer 2, Comment 8
Implementation between July 2020-February 2021 (line 99) and also mention as July 2020 to July 2021 (line 103).
Response: We apologize for this typo.
Changes:
(Line 111)
We extracted data from the pathology testing laboratory on the number of tests (HIV, syphilis, chlamydia, gonorrhoea) and test positivity for each clinic from two time periods: 1) Pre-implementation phase: January 2019 to June 2020; and 2) Post-implementation phase: July 2020-July 2021.
Reviewer 2, Comment 9
Lines 118-123: These statements about funding and availability of data can be removed. They are mentioned in declaration section.
Changes:
We have removed these statements.
Reviewer 2, Comment 10
Lines 127-128: “Pre-implementation, there was no significant change in the total chlamydia tests (p=0.246, Figure 1).” What was the period? Do you mean monthly change (per month)? Please indicate. This comment is also applied on line 140 and line 149.
Response: We have clarified this in the text.
Changes:
(Line 145)
Pre-implementation, there was no significant change in the mean monthly chlamydia tests (p=0.246, Figure 1).
(Line 174)
Pre-implementation, there was no significant change in the mean monthly gonorrhoea tests (p=0.826, Figure 2).
(Line 185)
Pre-implementation, there was no significant change in the mean monthly syphilis tests (p=0.122, Figure 3).
Reviewer 2, Comment 11
Figures 1, 2 and 3: label both phases on the figures (i.e. pre and post-implementation).
Changes:
We have added the labels to the Figures.
Reviewer 2, Comment 12
Training was provided between May and July 2020; but this period was also included in Figures 1-3.
Response: We apologize for the confusion. We have added further text to define what we mean by the post-implementation period.
Changes:
(Line 110)
We defined post-implementation in our study as after the month when clinics started seeing patients using the GP hub-and-spoke model.
Reviewer 2, Comment 13
Supplementary file: Supplementary Table 1 summarises the impact of the GP hub and spoke model before and after implementation. Impact on what? Please indicate.
Changes:
(Line 446)
Supplementary Table 1 summarises the impact of the GP hub and spoke model on knowledge, attitudes and practices before and after implementation.
Reviewer 2, Comment 14
STI or STIs. Please ensure consistency throughout the manuscript and supplementary file.
Changes: We have made changes to consistently use STI throughout.
Reviewer 2, Comment 15
Line 204: “Our model of care uses the nurse to do the initial assessment.” This is only mentioned here. This and additional important information about the model design, components and evaluation should be provided in methods section.
Response: We have added further details of the nurse-led model.
Changes:
(Line 344)
The nurse-initiated model, where the nurse completes the initial risk assessment, and then the patient sees the GP for clinical care, helps embed and strengthen the STI care patient pathway. Embedding the nurse in the care pathway is crucial to the sustainability of the model. There is high staff turnover in GP clinics and relying on one work group to maintain skills is a risk. There is an added benefit to workforce and career pathways for sexual health nurses in embedding this model in GP clinics.
Reviewer 2, Comment 16
Conclusions: A statement to recommend further evaluation can also be added here.
Changes:
(Line 302)
Further implementation research is warranted to evaluate its scalability in other settings and for managing other conditions.
Reviewer 2, Comment 17
Title: “a mixed-methods evaluation”. Nothing is mentioned in the text about the mixed methods. What does this mean?
Changes:
(Line 103)
We used a mixed-methods approach: extracting quantitative data on HIV/STI testing complemented by qualitative interviews to evaluate the acceptability and feasibility of integrating the hub-and-spoke model into each GP.
Reviewer 3 Report
The issues discussed in the article are of great importance for the prevention and treatment of STIs. The authors provide preliminary information showing the usefulness of the model for sexual health services, and the potential to use it in various countries. The subject of the article is closely related to the profile of the journal.
The article has been prepared in accordance with scientific requirements. Both the introduction and the methodological part do not raise any objections. The information provided is comprehensive (also thanks to its broader presentation in the appendix) and understandable.
My only comment concerns the analysis of qualitative research results. I suggest explaining briefly how the content analysis was carried out, and how the accuracy and reliability of the results were ensured. The results of this analysis are presented in the appendix but they are not easy to read. Please consider organizing them in a table, indicating the main topics, detailed topics, and quotes illustrating them.
Author Response
Reviewer 3, Comment 1
The issues discussed in the article are of great importance for the prevention and treatment of STIs. The authors provide preliminary information showing the usefulness of the model for sexual health services, and the potential to use it in various countries. The subject of the article is closely related to the profile of the journal.
The article has been prepared in accordance with scientific requirements. Both the introduction and the methodological part do not raise any objections. The information provided is comprehensive (also thanks to its broader presentation in the appendix) and understandable.
Response: We thank the reviewer for their time in reviewing our research and your positive comments.
Reviewer 3, Comment 2
My only comment concerns the analysis of qualitative research results. I suggest explaining briefly how the content analysis was carried out, and how the accuracy and reliability of the results were ensured. The results of this analysis are presented in the appendix but they are not easy to read. Please consider organizing them in a table, indicating the main topics, detailed topics, and quotes illustrating them.
Response: We agree with the reviewer and have added a reference and more details about the content analysis. We also re-organized Appendix 4 which now has a table that summarizes the main themes and subthemes. We then provide further details with clearer subheadings, including quotes that illustrate each theme and subtheme.
Changes:
(Line 129)
All interviews were recorded and data analysed using a content analysis approach to summarise the main themes.[17] Researcher (JO) identified initial themes, and reviewed and refined the themes with the research team. Transcripts were then re-read, and noteworthy phrases or concepts were coded and categorised into themes and sub-themes derived both deductively from the interview schedule and the data itself.
Round 2
Reviewer 2 Report
The manuscript has been improved and all my comments have been properly addressed.